# Engineering Students' Perception on Self-Efficacy in Pre and Post Pandemic Phase

Clara Viegas *, Natércia Lima and Alexandra R. Costa

CIETI/ISEP, Polytechnic of Porto, R. Dr. António Bernardino de Almeida, 4249-015 Porto, Portugal; nmm@isep.ipp.pt (N.L.); map@isep.ipp.pt (A.R.C.)
* Correspondence: mcm@isep.ipp.pt; Tel.: +351-228-340-500

**Abstract:** During 2020 and 2021, the world experienced a global change in everyone's daily lives due to the COVID-19 pandemic. Students were confined in their homes but, luckily, had access to online classes. This study aims to assess the changes in self-efficacy perceived by engineering students in a school in Portugal. By helping to understand how students have changed their learning capacities, developed new strategies, and/or need more (or different) support to learn, teachers can target their teaching methods accordingly and contribute to a more sustainable education. A questionnaire was constructed and validated to assess students' perceptions before and after the associated lockdowns. Five theoretically supported factors emerged from a statistical factor analysis: Communication and Empathy; Focus and Personal Organization; Teamwork and Individual Work Capacity; Technical and Cognitive Resources Management; and Emotional Resources Management. This work shows students' percept that they improved their teamwork and individual work capacity and their technical and cognitive resources management. In general, students seem to have been able to be more autonomous as they managed to work and develop their cognitive resources; however, their emotional state and ability to focus decreased. Perceived self-efficacy was less affected in older students than in younger ones, suggesting that this group may have adapted better to the pandemic restrictions. Students who were already at university showed less impact than those moving from high school to university. There was also a difference between those who endured these changes at only one level of education and those who endured them at both levels (high school and university), with this last group being the most negatively affected.

**Keywords:** students' perception; learning; higher education; students' academic competences; students' interpersonal competences; self-efficacy

## 1. Introduction

Few moments in human history have brought such rapid and marked changes as the COVID-19 pandemic crisis. In the field of education, thousands of students and teachers around the world have been placed at home using online platforms as schools. The long periods of confinement that took place between 2020 and 2022 isolated students from their social contacts, locking them in their houses. Their rooms became, at the same time, a bedroom, a place of study, and their classroom; these were the most fortunate. Some students went through greater limitations having to share space and resources with others or even not having access to the resources they needed, such as computers or the internet [1,2].

Many researchers have studied the effects of these lockdowns on students, revealing consequences in terms of sociability, lack of motivation and interest in class and learning, and even in terms of mental illness, namely increased levels of anxiety, stress, and depression [3–5].

After three years, the pandemic seems to have subsided, and gradually students are returning to the classroom. This return to normality will not, however, erase the

experiences lived during the pandemic, several months of confinement, and long periods of homeschooling. According to UNESCO, "In a post-COVID-19 world there will be a great need to cure the separations that have arisen due to quarantines and distancing restrictions. We will need to think creatively about ways to reconnect people. Trusting young people and empowering them to think and act together is one important way to accomplish this" [6] (p. 14). Lessons can be learned if we understand more clearly what the major changes were, some of them irreversible, and how to work upon this new scenario to build an even stronger education, perhaps using some strategies and resources developed during that time. Nor teachers nor students are the same after this predicament, so understanding what really changed in terms of students' learning might be a way to develop a more sustainable education.

Even though the pandemic period has been extensively studied in recent research, there are still some important issues that need our attention, namely how students perceive classes and their learning in this return to normality. Are students learning in the same way as before? Are they developing the necessary competences equally as before? Do students percept any change in his/her behavior or study habits? These are some of the questions that are not yet fully addressed in the literature and are fundamental to better understand students' reality. The problem addressed in this work aims to contribute to this better understanding of students' perceptions regarding this return to face-to-face teaching in terms of how the years in confinement have affected them. This way, teachers may better adapt to this new reality, adequate their teaching and methodologies or resources to enhance students' learning and develop the necessary competences.

This study was developed in a higher education institution, namely the school of engineering, with 487 participants through a survey. The primary goal of this work is to identify differences in students' behavior (compared to the prior pandemic era), namely interpersonal and academic competences. These competences may be influenced by students' perception of their self-efficacy. A secondary objective is to understand whether there are significant differences between students who undertook these restrictions mainly in high school and those who were already at university.

In addition to the introduction, this paper is organized into five sections and its aim is to study whether there are statistically significant differences between students' perceptions of their self-efficacy when comparing the pre- and post-pandemic phases. Section 2 is devoted to the effects of the pandemic on education, focusing on students and a theoretical background on students' perception of self-efficacy. In Section 3, the research methodology is lined up, and the reliability and validity of the study are addressed. Then, in Section 4, the results are presented and analyzed. Finally, Sections 5 and 6 discuss the results with the literature and draw conclusions that contribute to the research question: "*In which way the COVID-19 Pandemic with its lockdowns affected students' self-efficacy perception?*".

## 2. State of the Art

### 2.1. Students' Classrooms since the Pandemic

Between 2020 and 2021, the world lived through the COVID-19 pandemic, and most countries endured global lockdowns that lasted for months. Schools and universities developed online teaching during those periods. During the pandemic, teachers and students adapted to online conditions and, in several cases, reinvented their way of teaching and their way of learning. Some studies [7–10] indicate that teachers made an effort to diversify teaching resources and applied new methodologies and support. Other studies [11–13] also indicate that students struggled with online classes and felt that they were more productive when there was interaction between the teacher and the students or between students. Some students deeply felt the lack of contact with teachers and other students [5,14].

After enduring nearly two years with restrictions to face-to-face classes, higher education institutions returned to their normal functioning. According to Hess [15], despite all the experiences and everything that happened during the pandemic period, the majority of schools returned to their usual rhythms and routines, and little was changed. Teachers and

students, eager to return to the face-to-face mode, drop most of the online resources and materials developed during the pandemic [15]. Despite this absence of changes in terms of routines and in terms of school life, it is expected that such a long period of isolation will cause behavioral changes, particularly in young people, for whom social support from the group they belong to is of great importance. Therefore, it is important to study this return to face-to-face learning, namely by listening to students and understanding their perceptions of what has changed since the end of the pandemic period. Several research studies have documented how students reacted to this new reality during the pandemic, but few mention the modifications students felt in their academic and interpersonal competences. In fact, the researchers' focus on students' perception of the impact of the pandemic period focused on their opinions about the shift from face-to-face study to online study [11,16–18]. Little attention has been paid to the student's views on what has changed since their return to face-to-face learning, how they perceive the relationships with their colleagues and teachers, the productivity of their learning in general, and their emotional state after this period. One of the exceptions is the study developed by Becker et al. [19]. According to these authors, the impact of the long period of confinement on the students' skills may vary depending on the student's previous experience, that is, whether or not they were already attending the same degree prior to confinement [19]. This work is intended to contribute to the continued filling of this gap with the goal of addressing students' perceptions of themselves and their learning and how and what has changed.

### 2.2. Students' Perceptions

According to Curelaru et al. [3], "Perceptions are defined as complex mental processes by which people understand, interpret, evaluate, and form a picture of social phenomena" (p. 2). This interpretation of reality obviously affects the behavior of individuals. Thus, students' perceptions of how they were affected by the lockdowns may now, on their return to face-to-face classes, influence their behavior, attention, motivation, emotions, and satisfaction level.

Students' self-assessment is important as it represents a measure of self-efficacy that may influence students' behavior. Self-efficacy has been conceptualized as the belief that individuals have in their own ability to organize and execute the necessary actions to achieve certain goals [20]. Self-efficacy beliefs are, therefore, an important part of the motivational process, influencing the way the subject prepares for action. A positive belief in one's ability to perform a task can encourage behaviors that, by facilitating success, ultimately reinforce the belief in self-efficacy. Individuals with high levels of self-efficacy prefer to develop more challenging tasks and set more demanding goals for themselves; invest, at the same time, more in the tasks in which they are involved, showing greater levels of effort and persistence, overcoming more quickly the difficulties they face and maintaining focus on defined objectives [20–23]. Students' positive beliefs about their self-efficacy to manage academic tasks may also emotionally influence them by decreasing stress, anxiety, and depression [24]. By assessing students' self-efficacy beliefs, one can infer information about their predisposition to engage, invest and persist in learning activities.

Engineering students need to address several academic (subject-related) and interpersonal competences (soft skills) to fully cope with the profession when they graduate [25–27]. According to Alison Doyle [28], interpersonal skills are essential to the engineers' employability level and are considered equally important as content knowledge certificates. Doyle lists a set of skills she considers to be determinants for the success of professionals. In this list, she includes skills such as communication, empathy, leadership, active listening, conflict management, negotiation, positive attitude, and teamwork [28]. The importance of these skills is also covered by the CDIO (Conceive Design Implement Operate) initiative, which is a framework that defines standards for engineering degrees [29]. These standards divide the skills to be developed by future engineers into three major categories: technical, knowledge, and reasoning; personal and professional skills and attributes; and interpersonal skills. Comparatively, Hernández-March and collaborators [26], in a study of the skills

employers value in higher education graduates, organize these skills into four different groups: technical skills; interpersonal skills; cognitive skills; and methodological skills. The referred works were the starting point for defining the competences assessed by students' perceptions in this investigation.

In this study, perceptions were operationalized through students' self-assessment of a set of academic and interpersonal skills before and after confinement. This way, this work means to better understand how they feel affected by the years of lockdown.

### 2.3. Impact of COVID-19 on Students

In the last three years, many studies have focused on the impact of COVID-19 on young people and adolescents. Particularly with regard to students in higher education, research reports that the pandemic had an impact on several dimensions of students' lives, including their lifestyle, interpersonal competences, behaviors, emotions, feelings, and educational experiences [19,30–32]. Studies from different parts of the globe indicate some common issues experienced by students, such as a decrease in motivation due to social aspects and especially a lack of communication and interactions with teachers and peers [33–36]. In a systematic literature review carried out at the beginning of the pandemic crisis in 2020 on *The Impact of Social Isolation and Loneliness on the Mental Health of Children and Adolescents in the Context of COVID-19*, the authors found 83 articles that addressed this topic [37]. Sixty-three of these studies found a strong association between loneliness and mental health in children and adolescents, predicting that these problems may continue to arise for up to 9 years. One of the studies cited in the review states that children who had experienced forced isolation were five times more likely to need psychological support and to experience high levels of post-traumatic stress [38]. These results may indicate that it is expected that in the post-COVID-19 years, there will be a significant increase in mental health problems in young people. These results have been reinforced by a set of publications that, in recent years, have shown an increase in the prevalence of problems related to mental illness, such as stress, anxiety, lack of concentration, fear, sleep disorders, obesity, and depression [4,32,39–43].

Loneliness due to disease containment measures appears to be particularly problematic for young people, making them more vulnerable, namely because they feel deprived of the support of their peer group [37]. This may have more impact in educational settings where interpersonal relationships are more important [30].

In a study conducted by Ievers and collaborators on *The Impact of COVID-19 Restrictions upon Transversal Skills Development amongst Higher Education Students*, the authors found negative but also positive effects [44]. They mention positive impacts the developments that students report in their use of technology and digital literacy in general, which is corroborated by the findings of Gutierrez et al. [31]. In this way, with the right resources and support during the online classes, the authors argue students may even have the opportunity to improve their professional skills, such as communication, collaboration, self-efficacy, and digital skills. However, as these authors claim, these effects are likely to be due to students' exposure to online learning rather than to the lockdowns. Other positive impacts could be found in relation to citizenship, problem-solving skills, adaptability, self-reliance, and a small increase in inclination to listen to others and respect their point of view [19]. Still, other authors refer to COVID-19 as having a negative impact on students' problem-solving skills, time management ability, and teamwork, although their communication has been reinforced [19]. Ievers et al. [44] also mentioned a negative impact on the effective use of language, communication, the transmission of ideas, and confidence to engage in face-to-face communication right after the end of the pandemic restrictions. In fact, according to teachers' perception when returning to face-to-face classes, students had reinforced their communication not only with their peers but also with the faculty staff, including teachers (" . . . more friendly in terms of social interaction that any other semester I have had ever.") [19]. Other authors also mention a positive aspect (referred by students) of the lockdowns, which was that they were able to have more flexibility by living, working,

and studying in their cities and not having to commute anymore [33]. By changing some mindsets about how education must operate, society can emerge from this period with some positives to the learning process (using different and complementary ways of learning and practicing) and, on the other hand, help to reduce its ecological footprint in terms of transportation use, both contributing to sustainability [45].

## 3. Research Methodology

This study was conducted to better understand students' perception of some issues that might affect their performance not only in terms of learning the contents but also in developing social or technical competences. Its' main goal was to understand the cognitive and emotional effects COVID-19 had on students according to their own perceptions. The research question tackled in this work is: "*In which way the COVID-19 Pandemic with its lockdowns affected students' self-efficacy perception?*".

To accomplish it, a survey was applied through an online questionnaire (google form) and a mixed analysis of the quantitative and qualitative data provided [46].

The dimensions in the study were based on the work of Alison Doyle [28] and Crawley [29]. We started by outlining two major dimensions that we defined as Interpersonal competences and academic competences. The first was subdivided into the categories: communication, empathy, focus, organization, creativity, adaptability, and emotions/attitudes. The second dimension considered the categories: work capacity, technical proficiency, theoretical knowledge, and management. For each category, several questions were constructed to address each item from different angles. Finally, an open question was added to allow students' own reflections and to gather richer data about students' main concerns. Each question assessed two moments: the students' perception of their self-assessment in relation to these competences before and after the pandemic.

The questionnaire was first developed by the authors of this work and then validated through two focus groups [46] from the higher education school of engineering where this study took place. The first group constituted nine researchers, and the second of eight students. Both group participants presented different backgrounds, representing varied contexts, and were involved in diverse engineering degrees from different levels (post-secondary technical degree, major degree, and master's degree). Each question was discussed in terms of its clarity, the terms used for the Likert scale of agreement were also discussed, and at the end, the overall questionnaire was addressed. The inputs and suggestions each group provided allowed the researchers to make some adjustments to the initial questionnaire to make the purpose of each question clearer and perfectly understandable. Some questions were redrawn or combined.

The anonymous questionnaire ended up with the following:

- Two initial questions to characterize the population (participants' present age and school year (before the pandemic and present, that is, in the 2022/23 curricular year).
- Thirty-eight closed questions with the option "not applicable" and a 5-level Likert agreement scale (1—minimum; 5—maximum); students had to answer each of these questions considering their perception before the pandemic (the curricular year 2019/20 before March) and their actual perception (after the pandemic).
- An open question to welcome students' comments or final remarks.

All the questions were mandatory, except the open question and the question about the school year (by students' suggestion, as some of them, before the pandemic, were not in a formal educational situation and felt awkward about it). The questionnaire (Appendix A) took about 15 min to be answered.

The survey was delivered during the first semester of 2022/23 in a school of engineering, covering the students' community, involving students from different curricular years, including post-secondary, major's, and master's degrees. Teachers acknowledged students of the importance and relevance of the study and that the data collected would only be used for the purposes of this research. Students' anonymity was assured, as well as the voluntary nature of their participation.

A Google questionnaire was used in the process of collecting the answers, which link most teachers shared with students during a class. Other teachers shared the link with their students via email or on their course MOODLE page. These data collections occurred between 8 December 2022 and 15 January 2023, corresponding to the last weeks of the semester. The collected data were treated with the Statistical Package for the Social Sciences (SPSS) software [47].

The identification of the factors was driven by a factor analysis (FA) procedure using SPSS. FA is a data reduction technique used to group a large number of (observed) variables into a smaller set of representative factors. So, our theoretical categories were analyzed in terms of their consistency in the questionnaire. Each of the analyzed factors complied with more than one category. From the factor analysis, a total of five factors were identified: **F1**—Communication and Empathy; **F2**—Focus and Personal Organization; **F3**—Teamwork and Individual Work Capacity; **F4**—Technical and Cognitive Resources Management; **F5**—Emotional Resources Management. Each one is analyzed in two temporal periods: before the pandemic and the present time, that is, after the pandemic.

The reliability of the questionnaire and factors of analysis assessed its internal consistency. Table 1 summarizes the five analyzed factors, including the questions incorporated in each one as well as the Cronbach alpha [48] for each factor (before (b) and after (a) the pandemic) for 483 valid answers.

**Table 1.** Students' questionnaire internal consistency analysis.

| Factors | Questions (483 Valid Answers) | Cronbach Alpha | |
|---|---|---|---|
| | | Before (B) | After (A) |
| **F1**—Communication and Empathy | 1, 2, 3, 4, 5, 6, 7, 8, 9, 10 | 0.944 | 0.902 |
| **F2**—Focus and Personal Organization | 11, 12, 14, 15, 16, 17, 18, 19 | 0.943 | 0.922 |
| **F3**—Teamwork and Individual Work Capacity | 13, 20, 21, 22, 31, 32 | 0.937 | 0.888 |
| **F4**—Technical and Cognitive Resources Management | 23, 24, 25, 26, 27, 28, 29 | 0.929 | 0.891 |
| **F5**—Emotional Resources Management | 33, 34, 35, 36, 37 | 0.734 | 0.668 |

The former analysis shows internal consistency for all factors, although to a lesser extent for factor F5. To maintain this internal consistency at this high level, two of the 38 closed questions of the questionnaire were not used. In fact, these two questions were related to another category (creativity and adaptability) that theoretically did not fit into these five factors.

The qualitative analysis, related to the open question, was addressed using content analysis [46]. In general, students are not particularly attracted to this type of open question—it requires introspection, time, and effort—so when they answer, it is because they feel the need to express their feelings. The point was to identify the main ideas (besides different linguistic formulations) for each student and, when possible, relate them with the already represented factors. Each student's answer may express comments in more than one factor, so each answer may be spread out in several factors.

## 4. Data Analysis and Results

Of the 487 students who responded to the questionnaire, 483 were considered valid answers. Since the questionnaire was delivered to engineering students in a specific Engineering Institution, all respondents at this point (after the pandemic) are attending a college degree. Still, there were three students who answered "other" academic situation and one that answered "none", so these four answers were considered invalid.

### 4.1. Descriptive Analysis of the Collected Data

This valid sample constituted a diversified illustration in terms of age and in terms of academic background, and path, as illustrated in Figure 1.

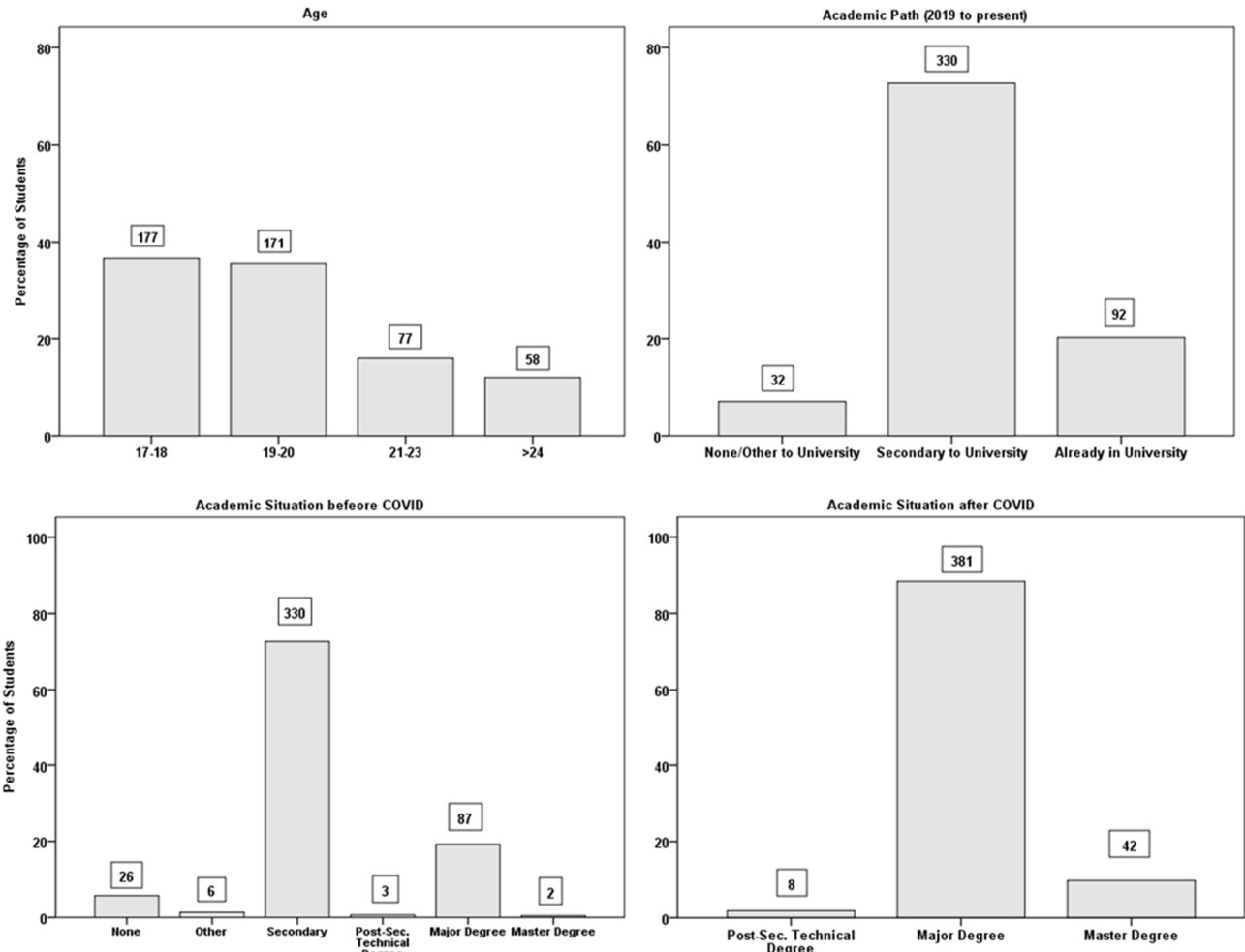

**Figure 1.** Sample characterization in terms of age distribution, academic path (**top**), and the academic situation before and after the pandemic (**bottom**).

In terms of age, the two larger groups had 17–18 years old (36.6%) and 19–20 years old (35.4%). The majority were students who were in high school at the time the pandemic broke out (68.3%). Some of them endured the pandemic for two years at high school, others for one year before entering university. Of those who endured the pandemic at university, 18.0% were in a major degree, 0.6% were in a post-secondary technical degree, and 0.4% attended a master's degree. Note that 5.4% were not studying at that time (that is, before the pandemic), and 1.2% were in "other" academic situations (which might mean nonacademic courses or non-degree courses). At the present time (after the pandemic), all students are attending a college degree. Most students were in a major degree (78.9%), some were already in a master's degree (8.7%), and others were in a post-secondary technical degree (1.7%).

The difference in the academic path (study level in 2019 until present) can influence students' perceptions since some moved from high school to university and may have experienced the pandemic period very differently from those who were already at university. So, to perceive these differences, the sample was split into three parts regarding their situation before the pandemic: the ones who did not attend school, those who were in high school, and those who were already at a university level.

Considering the 483 validated answers of each factor of analysis in both the pre- and post-pandemic phase (Table 2), the median value keeps unchanged in almost every factor, except for F2 (Focus and Personal Organization), where there is a downfall from 3.50 to 3.00.

**Table 2.** Median results in each factor of analysis for 483 valid answers (Likert scale 1–5).

| Factors of Analysis | | Median | Mean | Std. Deviation |
|---|---|---|---|---|
| F1—Communication and Empathy | Before | 4.00 | 3.73 | 1.171 |
| | After | 4.00 | 3.78 | 0.979 |
| F2—Focus and Personal Organization | Before | 3.50 | 3.35 | 1.148 |
| | After | 3.00 | 3.14 | 1.075 |
| F3—Teamwork and Individual Work Capacity | Before | 4.00 | 3.98 | 1.138 |
| | After | 4.00 | 4.16 | 0.882 |
| F4—Technical and Cognitive Resources Management | Before | 4.00 | 3.50 | 1.140 |
| | After | 4.00 | 3.79 | 0.952 |
| F5—Emotional Resources Management | Before | 3.00 | 3.22 | 1.216 |
| | After | 3.00 | 2.89 | 1.160 |

Even though the analysis of the mean value of each factor is not so relevant, the corresponding calculation of the standard deviation indicates a higher coherence of students in answering the post-pandemic questions than the pre-pandemic ones. This is totally understandable since students were asked to think back almost three years to respond to the pre-pandemic state.

### 4.2. Direct Correlations between the Factors of Analysis

After addressing the test of normality (Kolmogorov-Smirnov) and establishing the non-normality of the data, Spearman correlations were addressed between variables. Some differences occurred between the pre- and post-pandemic period (before and after in Table 3). Considering all students' valid answers, all factors show significant moderate correlations between them, being the lowest ones with F5. Addressing the correlations between factors at the same period (before/before or after/after) that appear in Table 3 in shadow, almost all correlations decrease its intensity (still being moderate) after the pandemic, the larger differences between F2 with F3 and F3 with F5 (boxes with border in Table 3). This might mean students felt these factors more apart after the pandemic.

**Table 3.** Correlation between factors before and after COVID-19 pandemic (B—before; A—after).

| Factors | Period | F1 | | F2 | | F3 | | F4 | | F5 | |
|---|---|---|---|---|---|---|---|---|---|---|---|
| | | B | A | B | A | B | A | B | A | B | A |
| F1 | B | | **0.514** ** | 0.649 ** | 0.239 ** | 0.615 ** | 0.443 ** | 0.557 ** | 0.360 ** | 0.505 ** | 0.177 ** |
| | A | | | 0.299 ** | 0.638 ** | 0.446 ** | 0.600 ** | 0.369 ** | 0.582 ** | 0.313 ** | 0.497 ** |
| F2 | B | | | | **0.388** ** | 0.590 ** | 0.438 ** | 0.512 ** | 0.323 ** | 0.519 ** | 0.222 ** |
| | A | | | | | 0.344 ** | 0.508 ** | 0.291 ** | 0.556 ** | 0.208 ** | 0.512 ** |
| F3 | B | | | | | | **0.797** ** | 0.651 ** | 0.473 ** | 0.481 ** | 0.262 ** |
| | A | | | | | | | 0.510 ** | 0.610 ** | 0.332 ** | 0.358 ** |
| F4 | B | | | | | | | | **0.623** ** | 0.460 ** | 0.246 ** |
| | A | | | | | | | | | 0.317 ** | 0.405 ** |
| F5 | B | | | | | | | | | | **0.486** ** |
| | A | | | | | | | | | | |

** Significant at the 0.01 level (2-tailed).

Analyzing the other correlations between factors before and after the pandemic stage (Table 3), other interesting correlations (from weak (<0.4) to strong (>0.7)) emerge. The first fact that stands out is the change in each factor between the two temporal moments (F1 before with F1 after, etc., indicated in bold in Table 3). Even though they are all significantly correlated, the highest ones are in F3 (strong correlation) and F4. This might mean that students felt less affected regarding Teamwork and Individual Work Capacity, and Technical and Cognitive Resources Management (in Table 2, it has already been shown that the median of these factors has not changed). Comparatively, the weakest one is found in F2 (weak correlation), which means that a greater percentage of students felt differently regarding their Focus and Personal Organization after the pandemic.

In relation to the correlations between factors at any temporal moment, a larger difference was found between F1 and F5, before and after (Table 3), which might mean that students percept these two factors (Communication and Empathy and Emotional Resources Management) further apart.

Considering the correlations between students' age and the factors, several very weak negative significant correlations appear before the pandemic (only F4 does not show this tendency) but no correlation after the pandemic. This could mean that the older the students were, the more confident they were in their capabilities. However, it also might mean that younger students really felt these factors stronger. Either way, these differences no longer appear after the pandemic. When analyzing the correlations between the factors and the study stage (secondary, post-secondary technical degree, major degree, master), no significant correlation appears, nor before nor after the pandemic.

### 4.3. Results Addressing Significant Differences Regarding Students Age and Study Path

After the descriptive analysis and the assessment of the correlation, the results were extracted to fully address the important issues under research. To perceive if there were statistically significant differences between the pre-pandemic stage (before) and the post-pandemic stage (after), a Wilcoxon nonparametric test was applied to the factors in the analysis.

First, an analysis was performed to understand if there were differences between the two moments in time (before and after). The results obtained for the 483 valid answers show (Table 4, first column) some significant differences: F2 and F5 decreased significantly, which means students felt negatively affected by the pandemic regarding their focus and personal organization and also in their emotional resources management. However, F3 and F4 show an increase. This means students percept a significant increase in relation to teamwork and individual work capacity and their technical and cognitive resources management. Only F1 does not show significant differences before and after the pandemic.

**Table 4.** Statistically significant differences in each factor for the total valid answers, by age interval and by study path.

| Factors | All Students (483) | Age of Students (Years Old) | | | | Study Path | | |
| --- | --- | --- | --- | --- | --- | --- | --- | --- |
| | | 17–18 (177) | 19–20 (171) | 21–23 (77) | >24 (58) | None/Other to University—*Largest Change* (32) | High School to University—*Large Change* (330) | Already in University—*Lesser Change* (92) |
| F1 | No difference | No difference | **Decrease** | No difference | **Increase** | Increase | **Decrease** | No difference |
| F2 | Decrease | Decrease | Decrease | Decrease | **Increase** | **Increase** | Decrease | Decrease |
| F3 | Increase | No difference | No difference | Increase | Increase | Increase | No difference | No difference |
| F4 | Increase | Increase | No difference | Increase | Increase | Increase | Increase | Increase |
| F5 | Decrease | Decrease | Decrease | Decrease | **Increase** | **Increase** | Decrease | Decrease |

However, these differences might be felt differently by distinctive groups of students, so the analysis of the significant differences was addressed by age and by their study path. This is also shown in Table 4, where the modifications (in comparison with the group of all students) are highlighted in shadow. For this analysis, the students were grouped by age intervals or by their study path. For 29 students, the information about the study path was

not available (in the analysis, was considered as missing data). So, the major results per factor are:

- F1: This factor is where the major differences occur in relation to the whole group of students. The group of 19–20 years old and the group who suffered the *largest change* in their study path (from high school to university) show a statistically significant change, considering their capacities in relation to communication and empathy decreased. On the other hand, older students have a different perspective, showing an improvement, which is also found in the group of students who suffered the *largest change* of all (the ones who were not studying or attended another type of education before the pandemic).

- F2: Only the older students and the ones who suffered the *large change* in their study path have a different perspective, considering their capacities in relation to concentration and personal organization improved.

- F3: Again, only older students (more than 21 years old) and the ones who suffered the *largest change* in the study path show the increase observed in the whole group, considering that their capacities in relation to teamwork and work capacity have improved, even though in this group the observed difference is more meaningful than in the whole group.

- F4: This factor is almost the same in every group, showing a significant increase in precepted capabilities regarding technical and cognitive resources management. Only in the group between 19–20, no significant difference between before and after the pandemic was found.

- F5: Again, all groups show the tendency to decrease their perception regarding their emotional resources management. Only the older group of students and the ones who suffered the *largest change* in their study path expressed an improvement after the pandemic.

It becomes clear that the older students and the group who suffered the *largest change* have similar perceptions. The overlap of these two groups represents 41% in relation to the age group (24 of these older students were part of the group that undertook the *largest change* in their study path).

The fact that these two groups increased their perception in all factors might indicate a difference in adulthood, more life experience, and, in the case of students who were not attending school, the increase of their capacities in factors related to the dimension of academic competences is understandable.

Students who have moved to a university level consider that they have only increased their perception of their capabilities in relation to technical and cognitive resources management (F4) and maintained the result of F3 (teamwork). This result might not be directly related to the pandemic but be a natural course of events in their lives due to the differences between the two worlds (secondary and university levels). It could also be related to students' lower level of maturity. Students who were already at university showed similar behavior apart from the fact that they did not show a significant change in F1 (communication and empathy), probably because they already knew their colleagues from college, and it was easier for them than for the previous group.

In terms of age, the group that shows a larger decrease in these factors' perceptions is the group between 19–20 years old (this group of students was in the majority the ones who entered university in the second year of the pandemic). Somehow this was not what was primarily expected. The younger ones (17–18 years old) were the students who endured the totality of the pandemic in high school. However, this result may indicate that students who endured the pandemic only in high school only experienced one kind of change. Students who experienced pandemic restrictions in both high school and university had to adapt to two ways of working during the pandemic. The results of the qualitative analysis obtained from the content analysis of the open question show support for some of the previous results, namely with factor F5. The overall analysis of the 30 answers (6.2% of the respondents) allowed us to identify some issues directly related to the questionnaire's main

goal. Table 5 shows these results summarized by groups, split according to their impact (negative, positive, neutral).

**Table 5.** Content analysis of the open question.

| Impact | Issues | Number of Students | Group Identification | |
| --- | --- | --- | --- | --- |
| | | | Age (Interval) | Study Path |
| Negative | Socialization difficulties (F5) | 3 | 2 in (19, 20), 1 in >24 | 1 missing; 1 none/other to university; 1 from high school to university |
| | Depression/anxiety (F5) | 4 | 2 in (19, 20), 2 in (21–23), 1 in >24 | 2 from high school to university, 1 already in university, 1 from major to master's degree |
| | Difficulty on focusing and self-discipline (F2) | 4 | 3 in (17, 18), 1 in >24 | from high school to university |
| | Anger and lack of patience (F1, F5) | 1 | (17, 18) | from high school to university |
| | Lack of confidence and motivation (F5) | 2 | 1 in (17, 18), 1 in >24 | 1 missing; 1 from high school to university |
| Positive | Personal organization (F2) | 1 | >24 | 1 already in university |
| | Resources (F4) | 2 | 1 in (17, 18), 1 in >24 | 1 missing; 1 from high school to university |
| Neutral | No changes | 5 | 2 in (17, 18), 1 in (19, 20), 2 in >24 | 2 missing; 3 from high school to university |
| | Difficulty in differentiate from other changes | 5 | 4 in (19, 20), 1 in (21–23), | 4 from high school to university 1 from major to master's degree |
| | Other comments | 2 | 1 in (19, 20), 1 in (21–23), | 1 missing; 1 from high school to university |

The most referred aspect was "*negative emotions*" which accounted for 11 answers; students expressed different emotions: "I felt compelled to interrupt my academic year due to depression/anxiety", " . . . had to take meds to control it", "nowadays I cannot focus for more than some minutes . . . ", "the major pandemic effect on me was the difficulties in socialization with colleagues . . . ". Considering the aspects that had a positive impact, two students generally refer to the pandemic helping them to learn in several ways or to use their own words, "they have learned from it". Other students are more precise and clearly identify the aspects the pandemic helped them to improve, as detailed in Table 5: "resources for remote work", "I can manage my time much better now . . . ".

Some students expressed they feel good and/or do not feel the pandemic has affected their behavior ("*no changes*"); some expressed it was difficult for them to understand the impact the pandemic had on their behavior as they have experienced a change in their study path and that modification also had an enormous impact in their lives. In "*other comments*", it was considered the feeling of gratitude two students have expressed for having the chance to be heard.

## 5. Discussion

Students perceived the pandemic restrictions differently according to their age and specific parameters. In relation to F1—Communication and Empathy, some authors point out that the young generation is used to communicating regularly online, even though the lack of face-to-face contact between peers could affect empathy. In this research, in relation to this factor, no significant differences were found in the student's assessment before and after the lockdowns, seeming to indicate that the isolation to which they were subjected did not impact on communication and empathy skills. In a way, these results go against

those obtained by Ievers et al. [44], according to whom (and also according to students' perspective), the isolation caused by successive lockdowns provoked a small increase in inclination to listen to others and respect their point of views [44]. It also contradicts the work of Becker et al. [19], in which teachers state students' communication ability clearly increased, being the students more friendly between themselves and with others.

Regarding factor F2—Focus and Personal Organization, the results show a significant decrease, indicating that students feel that they have lost skills related to the ability to focus on tasks, time management, and work organization. Indeed, many studies report consequences at the emotional level and on students' ability to concentrate, including time management issues [4,39–43]. Emotional disorders such as anxiety and stress may affect the subjects' ability to concentrate on tasks and, consequently, their focus and personal organization. Still, in relation to this factor, the older students and the ones who were not studying before the pandemic considered that their capacities in relation to focus and personal organization improved. This finding may be due to the fact that these students feel that they would be willing to make greater efforts to succeed in the face of such a significant change in their lives; or that these students, being more experienced and mature, may develop more positive self-efficacy beliefs. In contrast to the previous factors, students reported an increase in their competencies related to F3 and F4—Teamwork and Individual Work Capacity and Technical and Cognitive Resources Management, respectively. Our study showed a more significant increase in F4 than in F3, being the work capacity increase more pronounced in those who were not attending school during the pandemic or in the group of older students. These results are in accordance with other works that reported an enhancement in collaboration (teamwork), digital skills, and in self-efficacy [31]. Other studies also claim that the online environment favored some students who were more self-motivated or with higher self-regulating capacities [33,49].

In relation to F5—Emotional Resources Management, the literature supports that students' mental health has been severely affected, being one of the most cited factors. The reported social impact during the pandemic [32–34,36] might have left repercussions that students still need to overcome, such as anxiety, depression, sleep disorder or poor sleep. The findings of these works corroborate this study's results that there was a significant decrease in their emotional resources management.

The increase found in the group of students who were not in school during the pandemic may be consistent with Alexa et al. [33], who suggest students may have felt more flexibility in balancing their work and school lives. This might have been a reason for some of them to go back to school.

## 6. Conclusions

A better understanding of how students are currently coping with their learning, namely identifying possible gaps in their learning during COVID-19, deficits in the development of competences, or, on the contrary, the development of other skills or auxiliary tools, can indeed provide valuable information to teachers, who can thus better tailor their teaching to the student's needs. This more targeted teaching with a view to improving academic success can, in the long run, lead to more sustainable teaching.

With this work, some important aspects were more clearly identified, namely some positive aspects that teachers might consider continuing to use, such as auxiliary/complementary tools that allow students to practice more autonomously. However, these resources should be made available with an organizational plan and explanations of how and when to use them. This derives from the results pointing that globally, with the pandemic and confinement period, students percept they improved their teamwork and individual work capacity and their technical and cognitive resources management. However, their ability to focus and personal organization, as well as their emotional resources management, have deteriorated.

Students who were already at university show a smaller impact than those who have gone from high school to university.

Students who did not attend school before the pandemic and the group of older students (over 24 years old) show an improvement in their perception of their abilities in all factors, which means that this group is better able to cope with those restrictions, and some of them expressed they have learned from it. Students who were in high school before the pandemic underwent a greater change in their educational environment and probably other aspects of their life (when entering university, these students usually undergo changes in maturity, from being more independent, sometimes leaving home, changing city of residence, etc.), which might have influenced the change of perception observed in this group. This was also mentioned by some students, some of them clearly indicating this fact may have blurred their perception of the real impact of the pandemic. However, the only difference with the group of students who already were in university is strictly in relation to communication and empathy, where they show a decrease. Interesting evidence emerged differentiating students who had experienced the two years of the pandemic in high school from those who had experienced one year of restrictions and online classes in each level of education. The group who moved to the university showed more difficulty with communication and empathy issues, which may be due to the two major changes and adaptations they had to endure, one adapting to the online teaching and resources in high school and the other in university.

So, answering our research question, "*In which way the COVID-19 Pandemic with its lockdowns affected students' self-efficacy perception?*" this work shows that older students were less affected than younger ones, indicating a greater ability to adapt and cope with the pandemic restrains. In general, students seem to have been able to be more autonomous as they were capable of working and developing their cognitive resources, but their emotional state and ability to focus were reduced. All these factors influenced students' self-efficacy perception. Although students who changed their education level were the ones who suffered more, there was a significant difference between those who experienced these changes in teaching only at one level of education and those who experienced them at both levels (high school and university), with the former being the most negatively affected.

This study has some limitations, namely the fact that students were asked to answer the questions about their pre- and post-pandemic competencies in a single time point, with the presumed recall bias that this might introduce. Furthermore, the fact that the students are only from one educational institution, as well as the size of the sample, reduces the possibility of extrapolating these results. The fact that this study is based only on self-report measures is also one of the limitations of this study; moreover, given the age of some participants, it is impossible to categorically affirm if the evolution recognized by the results of this study is derived from the COVID-19 restrictions or the natural increase in maturity of those students.

To strengthen the consistency of these results, it would be interesting to carry out a longitudinal study that would allow monitoring of the evolution of these students' perceptions and possibly corroborate the results with an analysis of the teachers' perceptions of their students' learning.

**Author Contributions:** Conceptualization, C.V. and A.R.C.; data curation, C.V. and N.L.; formal analysis, N.L. and A.R.C.; methodology, N.L.; supervision, C.V.; writing—original draft, C.V., N.L. and A.R.C.; writing—review and editing, C.V., N.L. and A.R.C. All authors have read and agreed to the published version of the manuscript.

**Funding:** The authors would like to acknowledge the support provided by the Portuguese Foundation for Science and Technology Project, FCT UIDB/04730/2020.

**Institutional Review Board Statement:** Ethical review and approval were waived for this study due to ISEP/P.PORTO not having a formal Ethics Committee. To ensure ethical concerns, the survey was volunteer and anonymous. In the first section of the questionnaire, all participants were fully informed about the guarantee of anonymity, the research objectives, how the data would be used only for research purposes and the authors' contact details.

**Acknowledgments:** The authors would like to thank the students who completed the questionnaire and shared their insights. A special appreciation to the students and colleagues who participated in each focus group session and all the colleagues who delivered the questionnaire to their students.

**Conflicts of Interest:** The authors declare no conflict of interest.

**Appendix A. Questionnaire on Students' Perceptions Before and After the Pandemic**

This study aims to better understand possible changes felt by students after the pandemic period and the confinements they were subjected to. The idea is to compare their perception before the pandemic/confinements with the current situation (present time).

The data collected are anonymous and will only be used for scientific research. The results of this survey may be sent to interested parties by email to the authors.

The questionnaire has 38 questions and can be answered in about 15 min.

The scale of answers varies from 1 to 5, where 1 corresponds to completely false (not at all) and 5 completely true (yes, completely)

Thank you for your valuable contribution!

*Appendix A.1. Identification of the Profile*

Age:_____

Curricular year you attended/attend (before the pandemic/at present):

| | In 2019/2020 (1st confinement was in March 2020) | At Present Time |
|---|---|---|
| 10°ano | ☐ | ☐ |
| 11°ano | ☐ | ☐ |
| 12°ano | ☐ | ☐ |
| Ano Zero | ☐ | ☐ |
| 1°ano CTESP | ☐ | ☐ |
| 2°ano CTESP | ☐ | ☐ |
| 1°ano licenciatura | ☐ | ☐ |
| 2°ano licenciatura | ☐ | ☐ |
| 3°ano licenciatura | ☐ | ☐ |
| 1°ano mestrado | ☐ | ☐ |
| 2°ano mestrado | ☐ | ☐ |
| outro | ☐ | ☐ |
| nenhum | ☐ | ☐ |

*Appendix A.2. Questionnaire*

(an example of the answer format is presented, equal for the 38 questions)

| Before the Lockdowns: | not applicable | 1 | 2 | 3 | 4 | 5 |
|---|---|---|---|---|---|---|
| At Present Time: | not applicable | 1 | 2 | 3 | 4 | 5 |

*where 1 corresponds to completely false (**not at all**) and 5 completely true (**yes, completely**)*

1. I am able to participate actively in class
2. I am able to do work presentations
3. I am able to think and try to answer teachers' questions
4. I am able to communicate with colleagues in teamwork
5. I am able to defend my point of view in class
6. I am able to understand and/or argue with colleagues' points of view

7. I have the patience to be in a whole class and deal with classmates and teachers
8. I feel able to support colleagues with more difficulties
9. I get along easily with colleagues
10. I feel comfortable talking to teachers
11. I am able to manage my time
12. I feel motivated to work in class
13. I am able to make teamwork decisions
14. I am able to keep away from social networks during work periods
15. I am able to concentrate on the tasks I am doing
16. I am able to stay focused during a complete task
17. I am able to keep my attention when I am listening to others
18. I am able to keep focused on my studies
19. I manage not to be distracted in class
20. I can be open-minded to ideas different from my own
21. I can respect hierarchy and teamwork distribution of tasks
22. I can respect the ideas of my colleagues
23. I can easily write a text work
24. I can read, interpret, and understand an academic text
25. I can find ICT (Information and Communication Technology) tools that help me in my work
26. I can use and master the ICT tools I need
27. I can master varied resources to work on experimental concepts (e.g., simulators, remote laboratories . . . )
28. I can easily adapt to new situations
29. I can critically analyze information and/or the results of an assignment
30. I can be flexible in the way I work
31. I am able to create team spirit in teamwork
32. I can easily collaborate with my teammates
33. I am able to manage anxiety
34. I have little motivation to leave home for school
35. I am able to manage stress
36. I can balance my studies with my social life
37. In my daily life I often feel pessimistic and sad
38. I can be creative in my daily life

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
