# Peer review of "Engineering Students’ Perception on Self-Efficacy in Pre and Post Pandemic Phase"

_sustainability, doi:10.3390/su15129538_

Round 1
Reviewer 1 Report
The paper presents the analysis of the engineering students’ perception on self-efficacy in pre and 2 post pandemic phase. The topic is quite interesting and contributing, however, the paper needs some work before it can be published.
My first concern has to do with the instrument the authors used to measure interpersonal competences and academic competences. In page 6 they state that as reliability tests they only calculated Cronbach coefficients. However I believe that calculation is not enough. A factor analysis should be advisable to verify whether the dimensions are well identified.
Secondly, the questionnaire should be added as Appendix so that the readers could acknowledge the questions used.
Thirdly, in the Conclusion section authors should indicate the theoretical and practical implications of the study to demonstrate the contribution they made. Also, the limitations of the currents study and possible future research directions should be added.
Author Response
The authors are very grateful for your kind comments and important contributions to our work.
In the file uploaded you can find the answers to all the issues that you have mentioned. We hope this revised version answers all your suggestions.

Reviewer 2 Report
This study aims to investigate the perceptions of engineering students at a school in Portugal, about their own competencies before and after the pandemic. With that purpose, a questionnaire is validated. Results suggest that students perceive an increase in their self-efficacy related to team work and individual work capacity and technical and cognitive resources management, while there is a decrease in the competencies of focus and personal organization and emotional resources management, and these results are dependent on students age and changes in their education path, self-efficacy being better in older students and students that didn’t underwent any changes of educational setting (only high school or only university). The main problem is that students are asked to answer the questions about their pre and post pandemic competencies at a single time point, with the presumable retrieval bias, and even though, the survey is conducted at the end of 2022, a significant period of time having passed from both, pre and post pandemic.
The rationale could be improved. For example, I would like authors to justify in line 97 when they say that “Despite this absence of changes in terms of routines and in terms of school life, many teachers argue that they feel changes in their students' behavior.” Also, further development of why is important to know the changes that has taken place in students’ perceptions would be clarifying. Overall, the organization of the state of the art could be improved, at the present moment it doesn’t give a clear idea of what has the literature is that makes this study necessary. Finally, there is a mistake in the citation in line 183 and 202 “byliving”.
In the methodology section, nothing is said about the mandatory ethical approval required for any investigation. Information about it should be provided. In the description of the construction of the questionnaire, the resources in which is based and that are mentioned in the introduction, should be cited. Also, the questionnaire or some example questions should be provided. In the data analysis, there is a mistake in line 287 “(“. Nothing is said about how the participants where selected or what was the estimated sample size.
The presentation of the results is not clear. Some of them, the medians and correlations, are presented in the section data analysis, while the significance of the data is presented in the results section. As a consequence, the overall results are not clear, some data could be omitted due to their irrelevance or considered as a whole and not in separated sections, giving meaning and the relationships between them. The use of the word “former” in several occasions including the abstract, makes me confused about who were the students that improved, the ones that changed from high school to university or the other ones.
Discussion: mistakes in lines 446 “pont”, 468 “tomake”. In this study, there are some limitations as is usual the case. However, they don’t address any (cross sectional design, self-report measures…). The practical implications of the results should be addressed. Overall, the study can add some value to knowledge, although methodological limitations regarding the use of a cross sectional design to gather information about changes across time should be carefully considered when exposing the results and conclusions.
Author Response

(The authors gave the same response as above.)

Round 2
Reviewer 1 Report
A lot of work has been made, now the paper looks much better and higher quality!